# Ecological Responses of Meiofauna to a Saltier World—A Case Study in the Van Uc River Continuum (Vietnam) in the Dry Season

Hien Thanh Nguyen [1,2,*], Lucie Gourdon [2], Hoi Van Bui [2], Duong Thanh Dao [2], Huong Mai [2], Hao Manh Do [3], Thanh Vu Nguyen [4] and Sylvain Ouillon [2,5]

1   Faculty of Ecology and Biological Resources, Graduate University of Science and Technology, Vietnam Academy of Science and Technology, 18 Hoang Quoc Viet Str., Cau Giay, Hanoi 100000, Vietnam

2   Department of Water, Environment, Oceanography, University of Science and Technology of Hanoi, Vietnam Academy of Science and Technology, 18 Hoang Quoc Viet Str., Cau Giay, Hanoi 100000, Vietnam

3   Institute of Marine Environment and Resources, Vietnam Academy of Science and Technology, 246 Da Nang, Hai Phong 180000, Vietnam

4   Institute of Ecology and Biological Resources, Vietnam Academy of Science and Technology, 18 Hoang Quoc Viet Str., Cau Giay, Hanoi 100000, Vietnam

5   UMR LEGOS, IRD, CNES, CNRS, UPS, University of Toulouse, 14 Avenue Edouard Belin, 31400 Toulouse, France

\*   Correspondence: nguyen-thanh.hien@usth.edu.vn

**Abstract:** Increasing intensity of storms, typhoons, and sea level rise in conjunction with high water demand, especially for agriculture, in dry seasons in the Red River Delta may have led to seawater intruding deeper into the rivers' estuaries. Given that losses of agricultural productivity and shortages of freshwater resources are projected, a reliable early warning of salinity invasion is, therefore, crucially needed. To evaluate the impact of salinity variations on riverine ecosystems, distribution patterns of meiofauna were examined at 20 stations along the Van Uc River continuum in the dry season. Meiofaunal richness indices were higher in the estuary and slightly decreased upriver. Nematoda was the most dominant taxon in salty stations, while Rotifera was more abundant in the less salty ones. A multiple variate analysis showed a strong interplay among salinity, nutrients, and pore water conductivity, which shaped the meiofaunal distribution. The inclusion of pore water salinity, nutrients, and meiofaunal community structure indicated a greater extent of the saline ecosystem in the estuary, posing a greater risk of freshwater salinization. Our results highlight the potential role of meiofauna as bioindicators but also call for a reformation of salinity assessment for better freshwater conservation and management.

**Keywords:** salinity intrusion; meiofauna; community structure; bioindicator; ecotone

## 1. Introduction

River ecosystems are pivotal for humanity as they provide water for agriculture, industry, and power generation, as well as support high biodiversity values and many vital ecosystem services [1,2]. Nonetheless, given their strong economic potential, urbanization and industrialization rates in such river basins are often rapid, which increase competition for water resources and add more contaminants to aquatic environments [3]. Additionally, riverine ecosystems in low-lying areas are particularly susceptible to climate change, putting these habitats under serious threat [4]. Deterioration of water quality in both surface and groundwater are projected to be worsened in many regions, challenging the achievement of Sustainable Development Goals 6 (SDGs, clean water for all) and 11 (life below water). Hence, there is an urgent need for early warning systems and better management of rivers to provide multiple sustainable benefits [2,3].

As the third largest river in the Red River Delta, Vietnam, the Van Uc River has contributed to local livelihoods and the development of the city of Hai Phong, an important location in the northern key economic region of Vietnam [5]. Nevertheless, as with other rivers in the large coastal cities, the water quality degradation induced by both anthropogenic pollution and natural disturbance is a major concern to this densely populated river–sea continuum [6,7]. Nguyen et al. [8] emphasized a significant sea level rise trend in the Hai Phong coastal area, with a remarkably accelerated rate in the last 20 years (14.7 cm in comparison with 21.4 cm over 60 years). Seawater intrusion has recently been recorded moving further landward with longer retention [9–11]. Such events of extreme salinity intrusion in the northern region adversely impacted crop production and threatened freshwater resources [9,10,12].

It is worth highlighting that higher release of toxic heavy metals from the sediment to water environments, and increased heavy metal uptake by living aquatic organisms, have been observed, both induced by elevated salinity [13,14]. Additionally, as invertebrates with a freshwater affinity are sensitive to salinity change, their community structure will be altered by the intrusion of salt-tolerant or brackish species [15,16]. Those salt-tolerant species are poised to colonize new areas and expand their range, which can have severe impacts on ecological interactions and processes by altering the original components of the food web in a particular ecosystem [17]. The increase in salt concentration in the river, even in a small amount, has had results ranging from ecological mortality to sublethal effects, which, in turn, lead to biodiversity loss and reduction in related ecosystem services [18–20]. Hence, in the light of rapid population growth, increased demand for freshwater, and predicted climate change effects and biodiversity loss, a thorough assessment of salinity invasion impacts on the Van Uc River ecosystems is crucially needed to improve the city's risk mitigation and adaptation strategies.

The monitoring of salinity intrusion is often conducted using either direct water property measurements or remote sensing techniques [21]. Nonetheless, such approaches cannot reflect the integration of diverse environmental factors nor the long-term sustainability of river ecosystems [22]. In contrast, bioindicators are used to define the health of an ecosystem; they not only provide insights into their own response to environmental disturbance but also are capable of predicting how ecosystems might respond to future conditions [23,24]. Bioindicators are, therefore, among the proposed tools for monitoring ecological integrity and environmental pollution, with high applicability to water quality assessment [23,25]. Nonetheless, climate change impacts on freshwater environments are less well known compared to the terrestrial or marine realm [26]. Regarding salinity intrusion, numerous studies have focused on monitoring water quality changes and the hydrodynamics of both surface and groundwater in the saltwater-invaded area and its adjacent aquifer, yet few have addressed the response of organisms to saltwater intrusion [27,28].

Predominantly inhabiting the surface/groundwater interface, meiofauna, that is, small invertebrate species, have been widely used in many environmental quality assessments, owing to their benthic lifestyle, short generation time, and fast response to changing conditions [29–31]. Shifts in meiofaunal density, community structure, functioning traits, and other associated ecological indices have often been linked with or indicated for a wide range of environmental perturbations such as eutrophication, heavy metals, pesticides, seasonal variations, physical disturbance, and so on [31–35]. Importantly, the high sensitivity of meiofauna to salinity variations has been widely observed, including in polar, temperate, and tropical ecosystems [36–40], emphasizing their valuable role as a bioindicator for saline water invasion. However, only a few papers concern the composition of meiofaunal communities in Vietnam, and there remains a particular knowledge gap for meiofauna in the northern areas.

This study, therefore, aims to (i) investigate the changes in distribution patterns of meiofauna assemblages between different riverine habitats (upstream, freshwater/brackish ecotone, and downstream), and (ii) evaluate the shifts in community structure in relation to salinity variations along the Van Uc River.

## 2. Materials and Methods

### 2.1. Study Area

Located in the Red River Delta, the second largest of the important agriculture areas in Vietnam, the Van Uc River flows through the city of Hai Phong and finally meets the East Sea of Vietnam. The Van Uc River belongs to the Thai Binh River system and is one of nine distributaries of the Red River. It receives water and sediments inputs from both the Red River and the Thai Binh River via a complex network within the Red River Delta. The total river discharge through the Van Uc estuary was estimated at approximately $17.7 \times 109$ m$^3$/y (during the period 1989–2010), corresponding to 14.5% of the total water discharge from the Red–Thai Binh system into the Gulf of Tonkin [6]. The Van Uc sediment flux represents 14.4% of the total sediment flux of the Red–Thai Binh River to the Red River coastal area. Therefore, the Van Uc River is the third most important distributary of the Red River Delta in terms of water and sediment discharges (after the Day River and the Ba Lat River mouth). The Van Uc estuary is subjected to the Southeast Asian sub-tropical monsoon climate and experiences two distinct seasons, a wet season (May–October) and a dry season (November–April).

### 2.2. Sampling and Sample Processing

Samples were collected along the Van Uc River during the dry season in April 2021. Twenty sampling stations were selected following the salinity gradients, which presented downstream, brackish/freshwater ecotone, and upstream habitats (Figure 1, Table 1). It is worth noting that earlier in the dry season, the leading edge of saline water entering the river was estimated to reach 26–28 km upstream with a drastic change in salinity from brackish to freshwater occurred at 26 km (Figure 1, Km-26) from the sea [11]. In addition, the Van Uc River is characterized by a strong stratification of two distinct water masses during the dry season, with freshwater flowing seaward at the surface and seawater flowing landward near the bed [5]. Such typical estuarine circulation results from the combination of the longitudinal pressure gradient (a barotropic force, constant as a function of depth, and acting in a down-estuary direction) and the longitudinal density (salinity) gradient (a baroclinic force, increasing almost linearly with depth and acting in an up-estuary direction) [41]. Therefore, in order to track the salt front movement, the sediment samples were intensively taken both upriver and downriver from the station VU12 (Figure 1, Km-26) to assess the spatial changes in both environmental conditions and meiofaunal communities. The geographical coordinates and a brief description of the sampling stations are given in Table 1.

At each station, a Ponar grab sampler (sample area of $152 \times 152$ mm, at 4 m depth) was deployed three times to collect sediment. From the grab-sampling sediment, three subsamples for meiofauna and six others for environmental variable analyses, including granulometry and nutrient content, were then retrieved by inserting plexiglass corers (inner diameter, 3.4 cm, area 10 cm$^2$) down to 7 cm depth. All meiofauna samples were treated with 7% MgCl$_2$ to anesthetize organisms and then preserved in 4% formaldehyde solution while the other sediment samples were frozen until further analysis.

In the laboratory, the meiofauna samples were extracted by flotation with Ludox-TM50 (specific gravity of 1.18) and stained with Rose Bengal (0.5 g L$^{-1}$) before being enumerated and identified to the major taxon level under a stereomicroscope (EMZ-13TR, Meijitechno, San Jose, CA, USA). The following diversity indices were calculated for meiobenthos to assess their efficiency in describing environmental conditions [42]: the Margalef biodiversity (*d*), the Shannon index (*H'*), Pielou's evenness index (*J'*), and the Hill indices (*N1, N2*).

In addition, granulometric analyses were carried out in the laboratory using a Mastersizer 3000 (Malvern Panalytical, Malvern, Westborough, MA, USA). Organic matter in the sediment was estimated using the weight loss on ignition method and presented as the percentage of total matter (OM). Total phosphorous (TP) was measured using the vanado-molybdophosphoric acid colorimetric method, while total nitrogen (TN) was detected following the method of ISO 11464: 1994. Other environmental parameters, including

salinity and dissolved oxygen, were measured in the surface water, pH and temperature were measured in situ for both surface water (Hanna HI98194; Cluj, Romania) and pore water in the sediment, and electrical conductivity was measured to evaluate the salinity of pore water in the sediment (ECs; Field Scout Direct Soil EC Meter and pH meter; Spectrum Technology, Inc., Aurora, IL, USA).

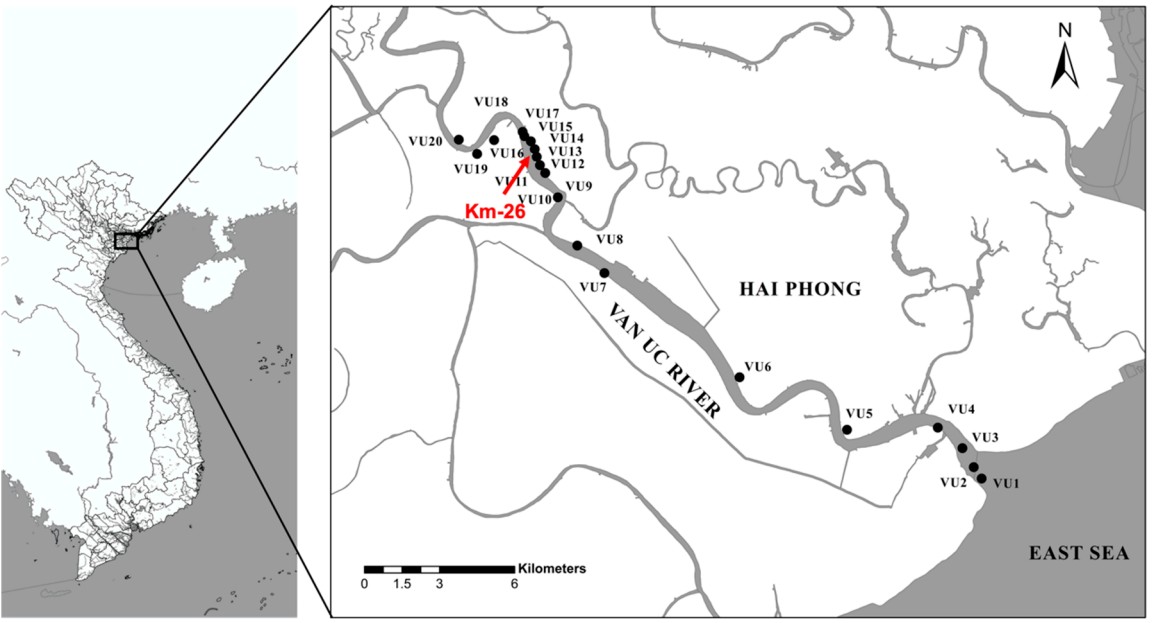

**Figure 1.** Map of the sampling sites along the Van Uc River.

**Table 1.** Description of sampling sites.

| Habitat | Station | Geographical Location | | Description |
|---|---|---|---|---|
| | | Latitude | Longitude | |
| Downstream | VU1 | 20.678319 | 106.700849 | Estuary, dense mangrove forest |
| Downstream | VU2 | 20.682121 | 106.698022 | |
| Downstream | VU3 | 20.688487 | 106.693951 | Mangrove forest, next to a freshwater outlet from agriculture irrigation channel |
| Downstream | VU4 | 20.695359 | 106.685127 | Scattered distribution of mangroves |
| Downstream | VU5 | 20.69464 | 106.652553 | |
| Downstream | VU6 | 20.712204 | 106.613992 | Close to livestock farm (pig) |
| Ecotone | VU7 | 20.747069 | 106.565513 | Close to rice field |
| Ecotone | VU8 | 20.756267 | 106.555787 | Close to rice field |
| Ecotone | VU9 | 20.77239 | 106.548795 | Close to rice field |
| Ecotone | VU10 | 20.772398 | 106.548794 | Close to polychaeta farm |
| Ecotone | VU11 | 20.780523 | 106.544285 | Close to rice field |
| Ecotone | VU12 | 20.783151 | 106.542376 | Close to rice field (26 km from the estuary) |
| Ecotone | VU13 | 20.785923 | 106.541231 | Close to polychaeta farm |
| Ecotone | VU14 | 20.788484 | 106.540525 | Close to rice field |
| Ecotone | VU15 | 20.791128 | 106.53912 | Riverbank under construction |
| Ecotone | VU16 | 20.792836 | 106.536795 | Close to rice field |
| Ecotone | VU17 | 20.794296 | 106.536181 | Close to polychaeta farm, on fertilized soil/mud in the dry season |
| Upstream | VU18 | 20.791533 | 106.525937 | Industrial zone |
| Upstream | VU19 | 20.786903 | 106.519853 | |
| Upstream | VU20 | 20.791663 | 106.513265 | Close to rice field |

*2.3. Data Analysis*

Principal component analysis (PCA) was applied to elucidate the interaction between the environmental variables. To determine the water suitability for irrigation use, electrical conductivity (EC) was classified into four classes according to Richards [43]: low salinity for irrigation purpose (EC < 250 μScm$^{-1}$), medium salinity (250 < EC < 750 μS cm$^{-1}$), high salinity (750 < EC < 2250 μS cm$^{-1}$), and very high salinity (2250 < EC < 5000 μS cm$^{-1}$). All environmental variables were normalized prior to the calculation of an environmental resemblance matrix. The spatial changes for each variable were tested by one-way PERMANOVA analysis (univariate, no transformation, Euclidean distance).

Differences among habitats were tested by a one-way ANOVA (analysis of variance, factor: habitat) performed on meiobenthic univariate variables (density, ecological indices). Homogeneity and normality of the dataset were checked using Kolmogorov–Smirnov tests. When required, the data were log (1 + x) transformed. A post hoc Tukey's test was applied when significant differences were detected by ANOVA.

To address the variations of meiofauna community structure, PERMANOVA and SIMPER analyses were performed. PERMANOVA tests were designed with one factor (habitat) and performed on a meiofaunal community structure resemblance matrix (no transformation, Bray–Curtis similarity). Pairwise tests were performed for the main factors and interactions when significant results were obtained. Monte Carlo tests were applied when the number of available permutations was <100. Similarity percentage (SIMPER) analysis was used to determine which taxa were responsible for observed differences between upstream and downstream stations.

The relationship between the environmental variables and the meiobenthic community structure was explored by carrying out a BIOENV analysis and a distance-based linear models (DistLM) routine [44] with forward selection of the independent variables and 999 permutations. Distance-based redundancy analysis (dbRDA) was then performed to visualize the model output from the DistLM, which showed the influence of environmental variables on meiofaunal communities.

All univariate and multivariate analyses were performed according to the procedures described by Clark and Warwick [45], using the PRIMER V6 software package [46] and the PERMANOVA+ add-on [44], except for the univariate analyses of meiofauna density and ecological indices (STATISTICA 7).

## 3. Results

*3.1. Environmental Variables*

The PCA revealed a strong salinity influence on the sample distribution pattern, with the first two principal components explaining 68.8% of the total variance (Figure 2). The first axis (PC1) was defined by pore water electrical conductivity, salinity, and dissolved oxygen and explained 44.1% of variance, while the second axis (PC2) was correlated with sediment grain size together with water pH and explained 24.7% of variance. Three groups of samples were formulated, representing three riverine habitats along a gradient of salinity (Figure 3), with the first one, the downstream/brackish ecosystem, characterized by oligohaline to mesohaline water and very high salinity for irrigation purposes (VU1–VU6), while the second group (ecotone) was defined by oligohaline water and high salinity for irrigation purposes (VU7–VU17). The last three stations represented upstream habitat (freshwater), although their ECs were still medium salinity for irrigation purposes. Univariate PERMANOVA (one-way, factor: habitat) analyses showed significant differences between the three habitats, except for in the granulometric analysis (D50). Salinity, pore water conductivity (ECs), dissolved oxygen (DO), and temperature significantly decreased from the estuary toward upstream, while nutrients (TN, TP, OM) were significantly higher in the ecotone and upstream habitats compared to downstream (Table 2).

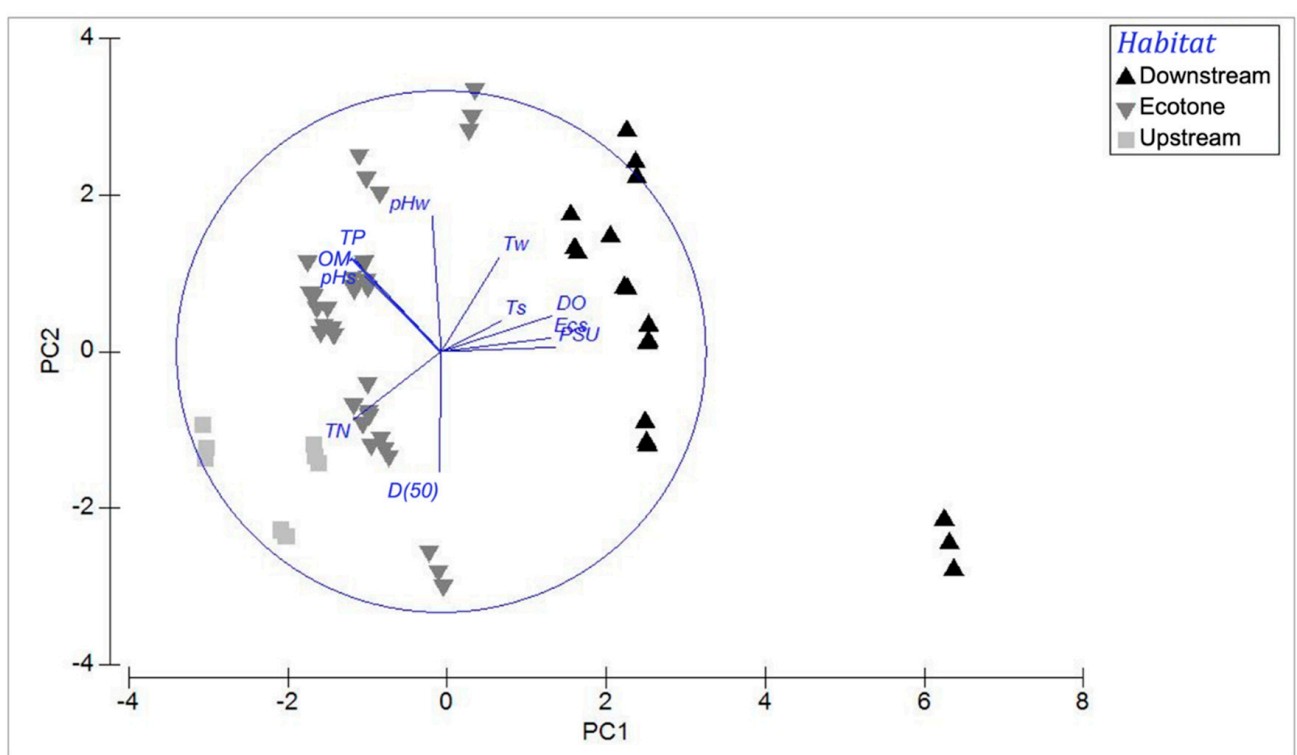

**Figure 2.** Principal component analysis based on environmental conditions in the Van Uc River (PC1 = 44.1% and PC2 = 24.7%). The circle displays correlation coefficients of the variables with the first two principal axes; the closer the projection vector is to the circle border, the higher the correlation is between the considered variable and its associated axis. $T_w$—water temperature, $T_s$—sediment temperature, DO—dissolved oxygen, $pH_w$—pH of water, $pH_s$—pH of sediment, $D_{(50)}$—median particle size, TN—total nitrogen, TP—total phosphorus, OM—organic matter, ECs—pore water electrical conductivity, PSU—practical salinity unit.

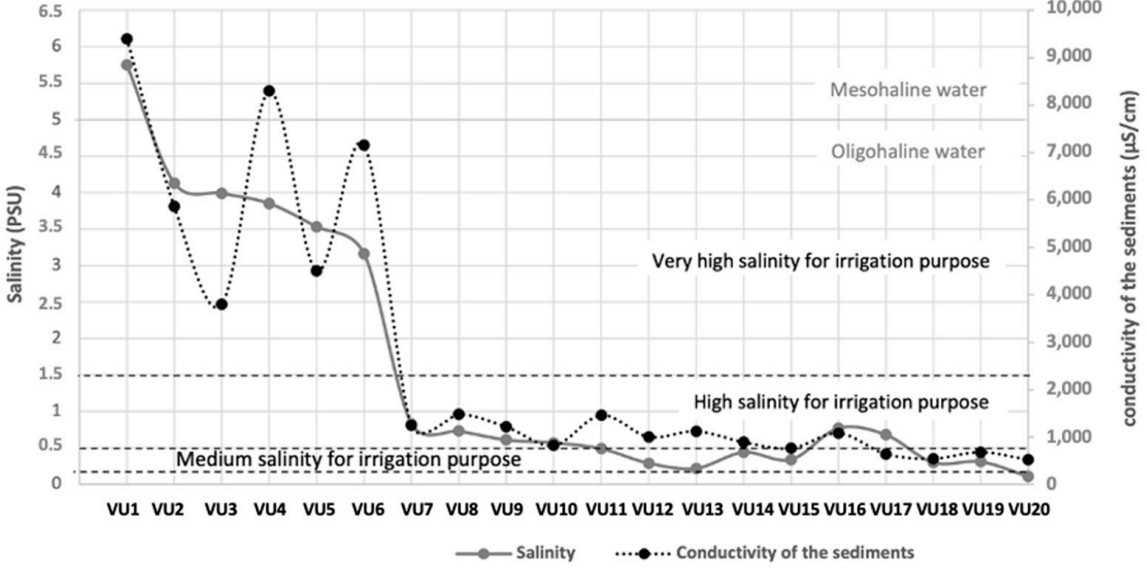

**Figure 3.** Salinity and sediment conductivity variations along the Van Uc River.

**Table 2.** Abiotic properties (mean ± SD, min.–max.) of sampling stations representing three habitats along the Van Uc River. $T_w$ — water temperature, $T_s$ — sediment temperature, DO — dissolved oxygen, $pH_w$ — pH of water, $pH_s$ — pH of sediment, $D_{(50)}$ — median particle size, TN — total nitrogen, TP — total phosphorus, OM — organic matter, ECs — pore water electrical conductivity, PSU—practical salinity unit.

| Abiotic Factor | Downstream | | Ecotone | | Upstream | | One-Way PERMANOVA | | | |
|---|---|---|---|---|---|---|---|---|---|---|
| | Mean ± SD | Min.–Max. | Mean ± SD | Min.–Max. | Mean ± SD | Min.–Max. | df | MS | Pseudo-F | *p* |
| PSU | 4.07 ± 0.84 | 3.16–5.75 | 0.54 ± 0.19 | 0.22–0.83 | 0.24 ± 0.1 | 0.11–0.31 | 2 | 27.272 | 348.8 | 0.001 |
| ECs ($\mu Scm^{-1}$) | 6503 ± 2048 | 3800–9400 | 1064 ± 270 | 640–1480 | 583 ± 75 | 520–680 | 2 | 24.8 | 150.39 | 0.001 |
| $T_s$ (°C) | 26.48 ± 0.49 | 25.8–27.1 | 26.23 ± 0.42 | 25.8–27.2 | 25.87 ± 0.43 | 25.4–26.4 | 2 | 5.0129 | 5.8345 | 0.007 |
| $T_w$ (°C) | 25.94 ± 0.19 | 25.7–26.2 | 25.90 ± 0.16 | 25.6–26.2 | 25.44 ± 0.02 | 25.4–25.5 | 2 | 4.9071 | 5.6221 | 0.04 |
| TN (%) | 0.72 ± 0.24 | 0.39–1.04 | 2.13 ± 0.79 | 0.48–3.52 | 2.49 ± 0.09 | 2.36–2.57 | 2 | 16.92 | 38.335 | 0.001 |
| TP (%) | 0.14 ± 0.02 | 0.10–0.16 | 0.16 ± 0.01 | 0.14–0.18 | 0.16 ± 0.0005 | 0.16–0.161 | 2 | 5.3827 | 6.3609 | 0.006 |
| OM (%) | 3.39 ± 0.45 | 2.81–3.99 | 4.04 ± 0.48 | 2.77–4.66 | 3.95 ± 0.63 | 3.32–4.74 | 2 | 7.7696 | 10.19 | 0.001 |
| $D_{(50)}$ ($\mu m$) | 49.21 ± 27.27 | 22–98.9 | 56.21 ± 22.44 | 27.7–109 | 60.07 ± 15.61 | 38.6–76.7 | 2 | 0.8199 | 0.8148 | 0.45 |
| DO ($mgL^{-1}$) | 5.59 ± 0.46 | 4.89–6.42 | 4.18 ± 0.23 | 3.88–4.73 | 3.22 ± 0.24 | 2.9–3.41 | 2 | 25.685 | 191.91 | 0.001 |
| $pH_w$ | 7.78 ± 0.1 | 7.54–7.92 | 7.81 ± 0.08 | 7.64–8.05 | 7.67 ± 0.06 | 7.61–7.77 | 2 | 7.2054 | 9.2108 | 0.001 |
| $pH_s$ | 7.2 ± 0.33 | 6.49–7.59 | 7.39 ± 0.12 | 7.2–7.72 | 7.38 ± 0.08 | 7.29–7.49 | 2 | 4.5527 | 5.201 | 0.016 |

### 3.2. Meiofauna Assemblage along the Van Uc River

The total meiofauna density (±SD) varied from 179 ± 27 inds/10 cm² (VU16) to 1454 ± 723 inds/10 cm² (VU14) (Figure 4). Higher variability of meiofaunal density was observed in the ecotone ecosystem compared to the downstream and upstream ones. Nevertheless, a one-way ANOVA test performed on the total meiofauna densities (under log transformation) showed that there were no significant differences among the three habitats (F = 2.482, *p* = 0.09).

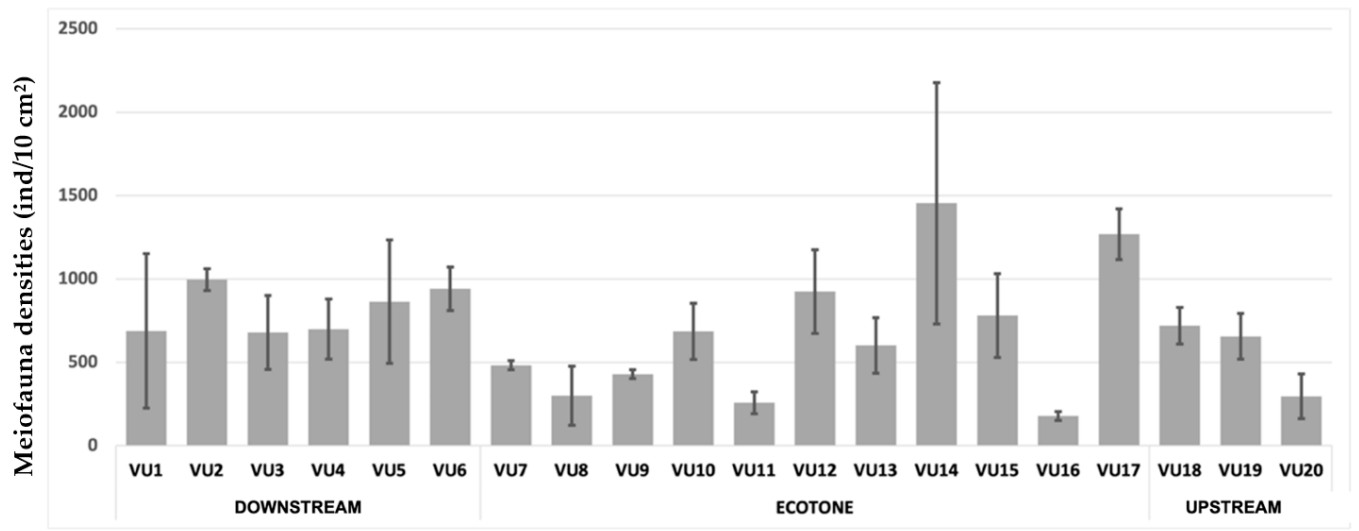

**Figure 4.** Meiofauna densities (mean ± SD/10 cm²) along the Van Uc River.

A total of 23 taxa were identified along the Van Uc River during the sampling period, with Nematoda the most dominant taxon (72.6%), followed by Rotifera (19.1%), Copepoda (2.2%), nauplii (1.1%), and others (4.1%). The "others" category contained all meiofaunal taxa whose percentage of representation was less than 1%, including Bivalvia (0.89%), Foraminifera (0.83%), Oligochaeta (0.74%), Polychaeta (0.6%), Amphipoda (0.55%), Turbellaria (0.35%), Ciliophora (0.33%), Ostracoda (0.13%), and Sipuncula (0.11%), together with some taxa presented at a very low density (less than 0.1%) such as Insecta, Kinorhyncha, Gastrotricha, Acari, Nemertea, Cumacea, Bryozoa, Isopoda, Gastropoda, and Tanaidacea.

The contribution of the different meiofaunal taxa present in the samples for each sampling site during the dry season is shown in Figure 5. Nematodes were always the most

abundant taxon, with the exception of VU14 and VU15, where Rotifera were predominant (with, respectively, 80.86% and 78.77%). Copepodes and nauplii were also represented by several individuals, changing, respectively, from 0.55% (VU13) and 0.32% (VU14) to 5.21% (VU9) and 5.52% (VU5). The representation of the "others" category changed from 0.33% (VU13) to 9.32% (VU20).

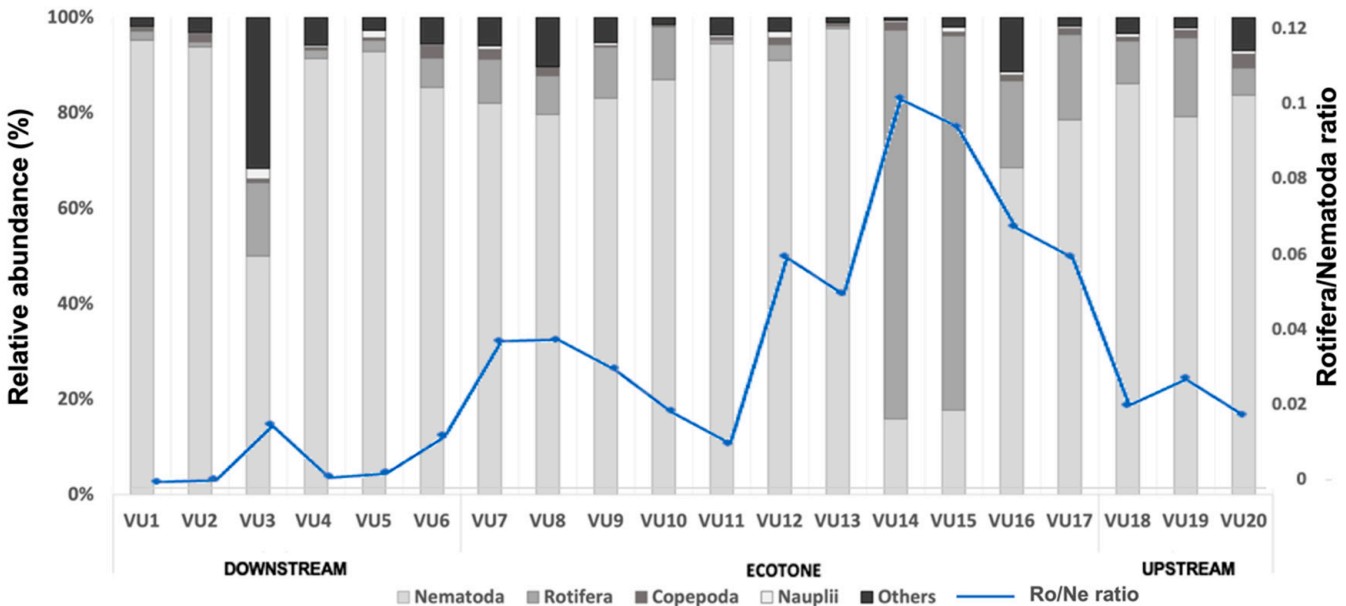

**Figure 5.** Proportional composition of meiofauna and Rotifera/Nematoda ratio along the Van Uc River.

The dynamics of nematodes and rotifers followed an inversely proportional relationship. When the percentage of Nematoda decreased, the percentage of Rotifera increased simultaneously and reciprocally. Moving inland, the Rotifera/Nematoda ratio in many stations increased as there were fewer nematodes, but more rotifers were found, especially in the ecotone, until nematodes took the lead again in the upstream habitat.

The significant changes in meiofaunal composition between stations were proven by the results of the one-way PERMANOVA test, which showed a pseudo-F of 3.03 and a $p$(perm)-value of 0.03 (Table 3).

**Table 3.** Results of one-way PERMANOVA test for meiofaunal community structure.

| One-Way PERMANOVA | | | | | | |
|---|---|---|---|---|---|---|
| **Source** | **df** | **SS** | **MS** | **Pseudo-F** | ***p* (perm)** | **Permutation** |
| Habitat | 2 | 4133.4 | 2066.7 | 3.0274 | 0.003 | 998 |
| Res | 57 | 38,911 | 682.65 | | | |
| Total | 59 | 43,045 | | | | |
| **Pairwise Test** | | | | | | |
| **Groups** | | | **t** | ***p* (perm)** | **perms** | ***p* (MC)** |
| Downstream, Ecotone | | | 2.1901 | 0.001 | 999 | 0.001 |
| Downstream, Upstream | | | 1.4951 | 0.032 | 999 | 0.056 |
| Ecotone, Upstream | | | 1.1445 | 0.236 | 999 | 0.26 |

In addition, the SIMPER routine demonstrated a high level of dissimilarity of VU14 from VU15, mainly contributed by Rotifera (72–81%) and Nematodes (16–24%) (Table 4).

**Table 4.** Results of the SIMPER analysis.

| Station | Average Similarity (%) | Taxa Contributions (%) | | | | | | | |
|---------|------------------------|-----------|---------|-----------|-------------|---------|----------|-----------|----------|
| | | Nematodes | Rotifers | Polychaeta | Turbellaria | Nauplii | Copepods | Amphipoda | Bivalves |
| VU1 | 60.85 | 95.24 | | | | | | | |
| VU2 | 92.88 | 95.86 | | | | | | | |
| VU3 | 73.88 | 93.23 | 1.56 | 1.48 | | | | | |
| VU4 | 68.43 | 73.42 | 17.07 | 1.98 | 2.74 | | | | |
| VU5 | 66.21 | 83.88 | 2.14 | | | 7.25 | 3.84 | | |
| VU6 | 75.45 | 80.79 | 10.33 | | | 2.58 | 2.88 | | |
| VU7 | 81.70 | 81.54 | 8.19 | | | | 3.24 | 3.74 | |
| VU8 | 63.80 | 91.18 | 5.44 | | | | | | |
| VU9 | 91.28 | 73.22 | 14.93 | | | | 5.54 | | 5.02 |
| VU10 | 74.75 | 89.51 | 7.94 | | | | | | |
| VU11 | 82.46 | 97.21 | | | | | | | |
| VU12 | 77.85 | 93.22 | | | | | 3.05 | | |
| VU13 | 80.13 | 99.14 | | | | | | | |
| VU14 | 65.11 | 23.95 | 72.93 | | | | | | |
| VU15 | 70.26 | 16.82 | 81.50 | | | | | | |
| VU16 | 79.76 | 75.75 | 12.09 | | | | | | 9.16 |
| VU17 | 88.69 | 90.98 | 2.99 | | | | 2.18 | | |
| VU18 | 82.48 | 93.34 | 2.53 | | | | | | |
| VU19 | 80.44 | 83.35 | 8.71 | | | | 4.12 | | |
| VU20 | 65.94 | 82.13 | 3.72 | | | 2.77 | 3.06 | 5.30 | |

### 3.3. Meiofaunal Ecological Indices

The meiofaunal richness (S) ranged between 6 in the ecotone (VU13) and 18 in the downstream habitat (VU6), showing a slight decrease from the estuary toward upstream (Figure 6). Being among the species richness indices, the Margalef biodiversity index (d) was also highest in VU6 (1.804) and lowest in VU13 (0.682), which concurred with the changes in the number of meiofaunal groups. The meiofaunal assemblage in VU13 was not only low in diversity but also unequally distributed, with nematodes predominant, making for the lowest Pielou's evenness index score (J') of 0.074 (VU13). The highest score for Pielou's evenness was obtained in VU16 (0.528). Furthermore, the maximum value of the Shannon index (H') was recorded at location VU3 (1.186), while the minimum value of this index was obtained at location VU13 (0.126). Concerning Hill's indices, N1 changed from 1.136 to 3.274, whereas N2 changed from 1.043 to 2.272. Both maxima were reached at location VU4.

One-way ANOVA tests performed on the different meiofaunal ecological indices showed that only the Margalef biodiversity index was significantly different among the three habitats, while the other four did not differ significantly. Results obtained from a pairwise comparison indicated that the Margalef index was significantly higher downstream compared to the ecotone habitat ($p < 0.05$), but there were no significant differences between the downstream and upstream, and ecotone and upstream, habitats.

### 3.4. Meiofauna in Relation to Environmental Variations

The marginal and sequential tests performed using the DistLM routine indicated that salinity, ECs, nutrients (TN, TP), sediment temperature, and sediment grainsize (D50) were environmental variables that significantly affected the community structure of meiofauna along the Van Uc River (Table 5, $p < 0.05$). The ordination plot generated via analysis of the linear model based on distance (dbRDA) revealed that 84.5% of fitted variation and 36.1% of total variation in meiofauna assemblages could be explained by the first two axes (Figure 7a). dbRDA1 represented 26.8% of the total variation and 62.7% of the fitted variation in the meiofaunal assemblages, showing a positive correlation with salinity and sediment temperature. D50, TN, and TP were correlated with dbRDA2, which explained 8.2% of total variation and 22% of the fitted variation. The first axis represents the variation of assemblages in relation to the spatial changes in salinity, with the right side of the

graph characterized by a brackish condition downstream with higher salinity. In contrast, ecotone samples were distributed on the left, with lower salinity values. The upstream samples were localized in the middle, but showed greatest similarity with the meiofauna in the ecotone. The second axis represents the spatial variation of meiofaunal assemblages in response to nutrient and sediment grain size changes. In our study, the increasing salinity was characterized by a greater diversity of meiofauna, with the predominance of nematodes, copepods, nauplii, amphipods, other shrimp-like crustaceans, and polychaetes (Figure 7b). In contrast, the left side of the graph, which presents ecotone and upstream habitats, is dominated by rotifers, insect larvae, some isopods, and acari. Nematodes were more abundant at the stations with higher salinity but lower nutrient contents, while rotifers favored the less saline condition and high OM and TN. Interestingly, copepods and nauplii mostly appeared in the direction of low TP and a coarser sediment grain size, whereas the insect larvae followed the same trend as rotifers.

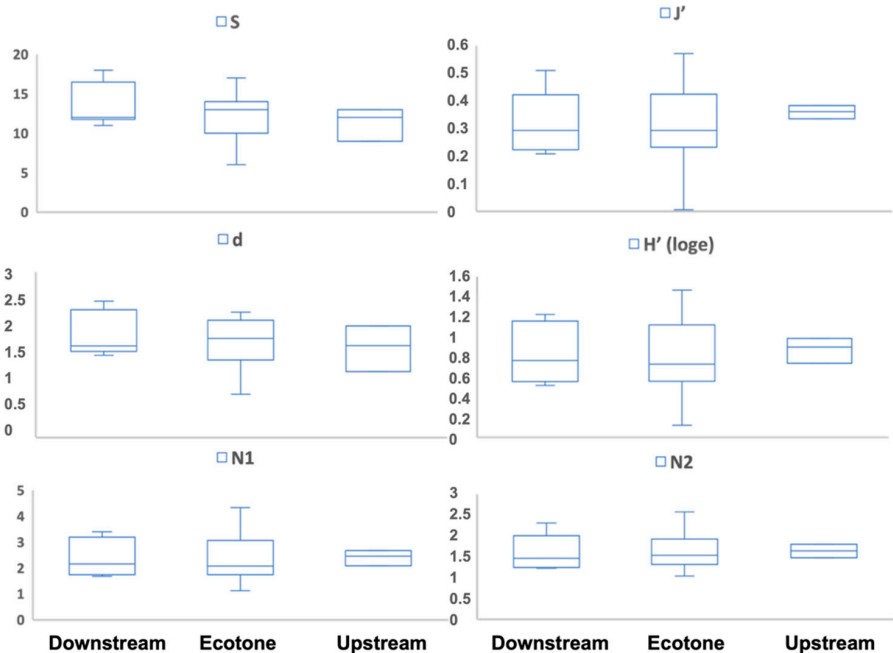

**Figure 6.** Meiofaunal diversity indices boxplots in downstream, ecotone, and upstream habitats along the Van Uc River in the dry season of 2021. Boxes display median, first and third quartiles, minimum, and maximum.

**Table 5.** DistLM analysis results showing correlations between environmental variables and meiofaunal community attributes along the Van Uc River. Values in red indicate significant correlations ($p < 0.05$).

| Marginal Tests | | | Sequential Tests | | | | |
|---|---|---|---|---|---|---|---|
| **Variables** | **Pseudo-F** | ***p*** | **Variables** | **$R^2$** | **Pseudo-F** | ***p*** | **Res. df** |
| pHw | 0.80628 | 0.483 | (+) pHw | 0.013 | 0.80628 | 0.459 | 58 |
| pHs | 0.8369 | 0.498 | (+) pHs | 0.025 | 1.5032 | 0.207 | 57 |
| DO | 2.2615 | 0.072 | (+) DO | 0.056 | 1.0279 | 0.409 | 56 |
| Ecs | 3.0077 | 0.03 | (+) Ecs | 0.084 | 1.7092 | 0.142 | 55 |
| PSU | 4.3192 | 0.01 | (+) PSU | 0.152 | 4.2724 | 0.01 | 54 |
| OM | 1.6501 | 0.171 | (+) OM | 0.173 | 1.3829 | 0.231 | 53 |
| TP | 1.466 | 0.235 | (+) TP | 0.228 | 3.6862 | 0.009 | 52 |
| TN | 1.7807 | 0.131 | (+) TN | 0.287 | 4.2498 | 0.008 | 51 |
| Tw | 0.35302 | 0.842 | (+) Tw | 0.299 | 0.82954 | 0.451 | 50 |
| Ts | 2.4743 | 0.064 | (+) Ts | 0.371 | 5.5701 | 0.001 | 49 |
| $D_{(50)}$ | 4.5717 | 0.006 | (+) $D_{(50)}$ | 0.428 | 4.7905 | 0.002 | 48 |

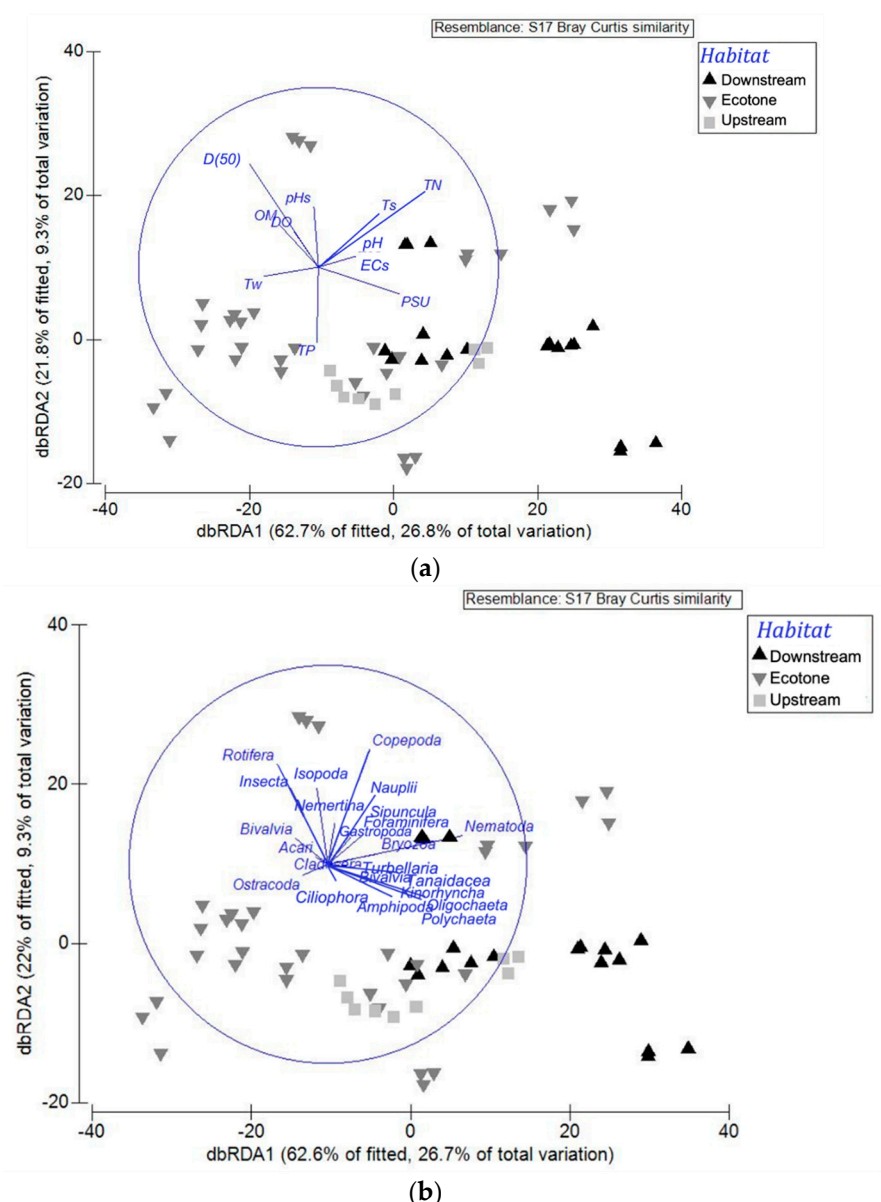

**Figure 7.** Distance−based redundancy analysis (dbRDA) of meiofaunal community responses to environmental variations showing (**a**) vector overlays of predictor variables and (**b**) vector overlays of species responses. The graph shows the effect of environmental variables on meiofauna community structure based on Spearman's rank correlations.

## 4. Discussion

Increasing salinization is a growing problem worldwide as it can heighten the stress faced by freshwater organisms and their mortality, thereby adversely impacting ecosystem functionality as well as the services and benefits to human societies that they provide [47]. Given species assemblages can shift to keep pace with climate change [48], gaining insight into the community structure transitions resulting from saltwater intrusion is crucial for managing and promoting coastal resilience [18]. In our study, the signature of saltwater intrusion was shifting of the meiofaunal community structure toward elevated pore water conductivity and surface water salinity along the Van Uc River.

It is widely accepted that one of the main factors influencing species distribution in estuaries is salinity [49–51]. Our findings are, therefore, consistent with several studies, which also identified salinity as an important independent factor determining meiobenthic communities' structure and describing total meiobenthic density and diversity changes [52–54].

Estuarine meiofauna tend to decrease in abundance and number of species as one moves from the sea to freshwater [55–57]. In our study, the meiofaunal richness indices (species richness and Margalef index) of the Van Uc River followed a linear relationship between species richness and salinity (Figure 6). A total of 23 taxa were identified during the dry season in the Van Uc River, with a higher number of taxa found in the brackish ecosystem, then slightly decreasing upstream. Our results revealed higher values compared to other estuaries elsewhere. Soetaert et al. [52] found 13 meiofaunal taxa in five European estuaries, while Pavlyuk et al. [58] recorded 11 taxonomical groups in the Cua-Luc estuary (North Vietnam). In the Mira and Mondengo estuaries, the meiofauna richness were 17 and 12 taxa, respectively [59,60]. Nevertheless, most of the studies in other estuaries were conducted in the intertidal areas with rather high levels of salinity, meaning the studies presented only the meiofaunal assemblage in the brackish ecosystem and neglected the communities in the brackish/freshwater ecotone as well as the freshwater communities. Exceptionally, in the Mekong Delta, where samples were collected across a wider range of salinity (0–25 PSU), a similar result was obtained with a total of 23 taxa recorded along 19 stations of the five estuaries [61].

Different from the richness indices, the meiofauna density in the Van Uc River was higher in the brackish ecosystem (downstream) in comparison with the freshwater ones (upstream), but the highest densities were observed at VU14 and VU17 (ecotone ecosystem) (Figure 4). Such fluctuation in the ecotone meiofaunal density could be related to the high environmental heterogeneity and/or the large amount of total nitrogen in the transitional area (Table 2). It is worth noting that agricultural activities (practices of tillage, rice crop fertilizing, and manuring for polychaeta farm) dominated along the two riverbanks during the sampling time, specifically from VU7 to VU20 (Table 1). Given that such anthropogenic sources can contribute greatly to river nutrient loading [62], the nutrients from the ecotone to freshwater ecosystems in the Van Uc River were considerably higher in comparison with other estuaries [29]. Interestingly, the same patterns of meiofaunal density were also observed at another five estuaries in the Mekong Delta [61], implying the importance of combined effects of salinity and nutrients on meiofaunal assemblages. Such interplay of several crucial environmental factors determining the discrepancy in the distribution of meiobenthic communities is well recognized at several estuarine benthic habitats worldwide [38,52,57,63,64].

In agreement with the above studies, our multivariate analysis revealed that the estuarine environmental gradients were strongly reflected in the meiofauna community structure, with salinity the greatest driving force, followed by TP, TN, and sediment grain size (Figure 7). Unlike the results of PCA, which categorized all stations into three groups based on environmental conditions alone, the distribution pattern of meiofauna assemblages in the Van Uc River during the dry season represented two distinct groups: the brackish and the merging ecotone–freshwater communities. Since meiofauna responses to different environmental variables are often highly species-specific [31], their structural parameters were found to be valuable indicators for detecting environmental changes, providing more pronounced effects on a taxon rather than on total meiofauna [65]. In our study, Nematoda was the most dominant taxon, which represented 72.6% of the total meiofauna density during the dry season. The same result was observed in a few studies worldwide [37,64]. The second abundant group of meiofauna was Rotifera, representing 19.2% during the dry season. This result is different compared to previous studies in other estuaries, where Copepoda was recorded as the second most abundant group [52,66,67], or Sarcomastigophora [68], Polychaeta [69], Tardigrada [70], or even Turbellaria [71].

As meiofauna respond differently to environmental variations depending on their functional traits and life strategy, the occurrence of nematodes and rotifers changed remarkably when there were salinity variations in the Van Uc River. More nematodes were found in the saltier stations, while rotifers preferred less salty ecosystems (Figure 7). The shifts in meiofaunal community structure occurred at VU14 and VU15, which exhibited the minimum amount of Nematoda and maximum density of Rotifera. The same meiofaunal

responses were observed at VU3, the brackish station that received freshwater discharge from the irrigation channel. In these three stations, when a sudden reduction in ECs occurred, a large proportion of Nematoda was replaced mainly by the rotifers and some other meiofaunal groups. Rotifer density then decreased at VU16 and VU17, which could have been in correlation with the elevated EC values at these two stations, implying sensitive responses of meiofauna toward salinity variations. These results concurred with the study of Majdi et al. [72] on meiofauna in the sediment of two headwater streams, Ems and Fulbach, in Germany. Ems, which was lower in EC values, was predominantly demarked by rotifers, with much fewer nematodes, while in Fulbach, with higher conductivity, nematodes took the lead, followed by rotifers and other groups. Such drastic change in meiofauna composition pairing with the fluctuation of pore water conductivity emphasizes the important role of electrical conductivity/salinity in regulating riverine ecosystems.

Interestingly, at the brackish and marine-influenced ecotone stations (VU1–VU13), the composition of Rotifera mainly consisted of marine ploima rotifers, whereas from VU14 toward freshwater-influenced stations, freshwater bdelloid rotifers rapidly increased. This calls for further research on what happened to the nematodes and other organisms. Comparing to organisms such as copepods and cladocerans, rotifers are more opportunistic, mainly due to their high reproductive rate. In many cases, they can quickly respond to environmental stresses, showing high sensitivity to elevated salinity [73,74], which makes them a valuable environmental indicator [75,76]. This result indicated that the leading edge of salinity intrusion was penetrating further landward during the late dry season compared to the estimation of salinity intrusion in the study by Nguyen et al. [11], which was conducted in an early month of the dry season. Our findings, therefore, highlight the potential risk of salinization as the water quality at most stations was classified from medium salinity to very high salinity for irrigation purposes during the dry season.

Along with salinity, nutrient enrichment is recognized as another very important factor influencing the meiobenthic taxa composition and abundance patterns [77,78]. In our study, the low density of Copepoda could have been related to the high nutrient contents (TP, TN) and organic matter in the sediment. Additionally, the overall low dissolved oxygen in the river could also have reduced this group as harpacticoid copepods are the most sensitive meiofauna taxon to low oxygen concentrations [79]. In contrast, the substantial decrease in dissolved oxygen (DO < 4 mgL$^{-1}$) in the upstream ecosystem (VU18–VU20) seems to have adversely impacted the rotifers and other meiofaunal groups, leaving a high percentage of nematodes, owing to their high anaerobic capacity [80]. The role of dissolved oxygen in structuring the benthic meiofauna was evidenced previously in several studies [81–83]. Erikson et al. [84] obtained the same results, showing strong interactions between nutrient inputs and oxygen depletion in tropical lowland rivers. Our findings suggest that the Van Uc River is not only subjected to salinity intrusion but also highly sensitive to pollution by nutrients and organic matter, with substantial impacts on meiofaunal community composition. However, intensive fertilizer use can increase soil salinity [85], which calls for salinity assessments to be reformed to better evaluate the freshwater salinization in the Van Uc River, induced by either the movement of the seawater or excessive fertilization, or both.

## 5. Conclusions

The meiofaunal community in the Van Uc River was characterized by high abundance and diversity. A total of 23 taxa were identified, with Nematoda the most dominant taxon and playing an important role in controlling the characteristics of the meiofauna assemblages, followed by Rotifera, Copepoda, nauplii, and other groups. Meiofaunal richness indices were higher in the estuary and slightly decreased upriver. In addition, the estuarine gradients were strongly reflected by the meiofaunal community structure, with salinity the greatest driving force, including salinity in the water column and pore water salinity (represented as electrical conductivity) in the sediment. The dynamics of meiofaunal density and community structure were best determined by the interplays among salinity, nutrients, and

dissolved oxygen. Both meiofaunal responses and environmental parameters of the Van Uc River indicated a high risk of salinization coupled with pollution induced by nutrients and organic matter. Further studies on nematode and rotifer interactions are needed as they may serve as a potential indicator for salinity intrusion assessment in long-term studies, especially in the context of climate change, in the future. Finally, it is also recommended that both natural salinity intrusion processes and anthropogenic-induced salinization should be considered, to gain insights into the dynamics of the salt front movement, which will support the development of better mitigation and adaptation strategies.

**Author Contributions:** H.T.N.: Conceptualization, Funding Acquisition, Methodology, Investigation, Formal Analysis, Writing—Original Draft Preparation; Writing—Review and Editing. L.G.: Investigation, Data Curation, Visualization, Writing—Original Draft Preparation. H.V.B., D.T.D. and H.M.: Methodology, Investigation, Data Curation. H.M.D. and T.V.N.: Data Curation, Writing—Review and Editing. S.O.: Writing—Review and Editing All authors have read and agreed to the published version of the manuscript.

**Funding:** This research was funded by the Graduate University of Science and Technology (GUST), part of the Vietnam Academy of Science and Technology (VAST), grant number GUST.STS.ĐT2020-ST02.

**Data Availability Statement:** Data are available from the corresponding author on request.

**Acknowledgments:** The authors gratefully acknowledge the International Joint Laboratory LOTUS— "Land Ocean aTmosphere regional coUpled System"—and the French National Research Institute for Sustainability Development (IRD) for their support.

**Conflicts of Interest:** The authors declare no conflict of interest.

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
