# Peer review of "Ecological Responses of Meiofauna to a Saltier World—A Case Study in the Van Uc River Continuum (Vietnam) in the Dry Season"

_water, doi:10.3390/w15071278_

Round 1

Reviewer 1 Report (New Reviewer)

This is generally a solid and interesting work concerning distribution of meiobenthos in a channel of the Red River delta from about a mouth upstream. The work may be treated as especially important since it is done in an agricultural region where the water quality deteriorates because of rising sea level. The study has been conducted with account of main milieu parameters and using appropriate data analyses.

I have only a few particular remarks.

Such indices as Shannon-Wiever, Pielou et al. are used mostly for assessing species diversity. In frame of this study, species identification or even sorting in provisory morphospecies were not conducted. Then what are elementary units for assessing diversity – major taxa such as Nematoda, Copepoda, etc.? It is necessary to explain.

Fig. 1. Separated square on the map of Vietnam does not correspond to the enlarged are with Van Uc River.

I don’t understand quite clearly the Vietnamese names – what is family name or given name. Seemingly, first author of the MS is cited as first author in References different in ways– as Hien, N.T. [10] and Nguyen, Thi Hien, right? It should be corrected.

Author Response

Response to reviewer 1

Point 1: Such indices as Shannon-Wiever, Pielou et al. are used mostly for assessing species diversity. In frame of this study, species identification or even sorting in provisory morphospecies were not conducted. Then what are elementary units for assessing diversity – major taxa such as Nematoda, Copepoda, etc.? It is necessary to explain.

Response 1: We agree with the reviewer that biodiversity indices are often used for assessing species diversity. However, regarding meiofaunal studies, particularly the ones related to environmental monitoring and ecological status assessment, numerous studies have used such indices as environmental indicators based on major taxonomic group of meiofauna (for examples, Moreno et al., 2008; Cui et al., 2021). In our study, we applied the same approach to calculate the biodiversity indices. Example published studies are provided below:

Moreno, M., Vezzulli, L., Marin, V., Laconi, P., Albertelli, G., & Fabiano, M. (2008). The use of meiofauna diversity as an indicator of pollution in harbours. ICES Journal of Marine Science, 65(8), 1428–1435. https://doi.org/10.1093/icesjms/fsn116

Cui, C., Zhang, Z., & Hua, E. (2021). Meiofaunal Community Spatial Distribution and Diversity as Indicators of Ecological Quality in the Bohai Sea, China. Journal of Ocean University of China, 20(2), 409–420. https://doi.org/10.1007/s11802-021-4550-5

Point 2: Fig. 1. Separated square on the map of Vietnam does not correspond to the enlarged are with Van Uc River.

Response 2: The suggested correction has been made.

Point 3: I don’t understand quite clearly the Vietnamese names – what is family name or given name. Seemingly, first author of the MS is cited as first author in References different in ways– as Hien, N.T. [10] and Nguyen, Thi Hien, right? It should be corrected.

Response 3: Thank you for your remark, Hien should be the first name, while Nguyen is the family name. The suggested correction has been made.

Reviewer 2 Report (New Reviewer)

The authors should consider a change in the title as I believe there are estuarine conditions, since the environment is more of a mesohaline environment.

Author Response

Reviewer 3 Report (New Reviewer)

This is an important paper focusing on the potential influence particularly of salinity for the composition, diversity, and distribution of meiofauna in estuarine ecosystems. The authors demonstrate that among the 23 detected meiobenthic major taxa, nematodes dominated the more haline areas, whilst rotifers increased in areas with decreasing salinity. Gradients of ecological variables (e.g. salinity, nutrients) were clearly reflected by the overall meiofaunal composition. Especially Nematoda and Rotifera and their interactions seem to be promising indicators for monitoring effects of (anthropogenically induced) changes of (a)biotic variables in riverine and estuarine habitats.

The manuscript is generally very well-written and comprehensible, as far as I am concerned (no English native speaker). However, I detected several potential slips of the pen throughout the manuscript. Moreover, partially also the syntax might be improved (e.g. placement of definite and indefinite articles).

The sections (Abstract, Mat. & meth., Results, Discussion) are absolutely adequate. The argumentation is comprehensible, and the conclusion is adequate, as far as I am concerned.

I suggest an improvement of the graphs shown in Figs 2 and 7, as they are somewhat diffuse, and the integrated terms are hardly readable.

The reference list seems, in my opinion, to be complete and appropriate.

I provide some detailed comments in an uploaded revision of the manuscript.

Apart from the above formulated critics, which demand minor revision, the manuscript is absolutely worth of being published in Water.

Author Response

This manuscript is a resubmission of an earlier submission. The following is a list of the peer review reports and author responses from that submission.

Round 1

Reviewer 1 Report

Dear Editor,

I revised for the first time the manuscript entitled "Meiofauna Responses to a Saltier World – The Case Study in Van Uc River Continuum (Vietnam) in Dry Season" by Hien et al.. The paper aims to assess how a salinity gradient affects the meiofaunal composition of the Van Uc river in the dry season. The results highlighted a nice shift between rotifers and nematodes. The topic is relevant in the prospect of future research because salinity and climate change might represent a destructive combo for freshwater meiofaunal assemblages in the following years.

I found the introduction and discussion well reasoned and structured, while I have several concerns about how the results have been presented. There are some aspects that must be clarified and improved before publication. Therefore, I suggest accepting the manuscript following major revisions. However, I want to highlight that the revisions are intended to improve the soundness and presentation of the results. I enjoyed the paper, and I think that it is worth paying a bit more attention to details to improve its quality.

I am not a native English speaker so I am not entitled to revise the language.

I am afraid that all figures are in poor resolution. Please, check that they are provided with 300 dpi.

Introduction

The aims of the study are presented at the end of the paragraph. However, the reader cannot understand point a). The three ecosystems wshould be indicated and well explained here. Please, revise this point to provide more information.

Lines 49-52: I suggest also considering one of the following studies:

Di Lorenzo T., Fiasca B., Di Cicco M., Cifoni M., Galassi D.M.P., 2021. Taxonomic and functional trait variation along a gradient of ammonium contamination in the hyporheic zone of a Mediterranean stream. Ecological Indicators, 132: 108268. https://doi.org/10.1016/j.ecolind.2021.108268

Cifoni M., Boggero A., Rogora M., Ciampittiello M., Martínez A., Galassi D.M.P., Fiasca B., Di Lorenzo T., 2022. Effects of human‑induced water level fluctuations on copepod assemblages of the littoral zone of Lake Maggiore. Hydrobiologia, 849, 3545–3564. https://doi.org/10.1007/s10750-022-04960-3.

Lines 62-63: please, reword because it is not clear for a non-native English speaker.  

Line 78: please, change to [19-23]. Please, revise accordingly throughout the manuscript when necessary.

Study area

Lines 100-114. The paragraph is well presented; however, it lacks a visual reference. If possible, I suggest adding some of this information in Fig. 1.

Lines 117-122. I did not understand from here which and how many stations are included in the marine brackish ecosystems and so on. Please, revise by providing this information. In my opinion, it is also necessary to add this information in Fig. 1, by coloring the three groups of stations, for instance. Finally, the figure’s caption needs to be more detailed by describing all the elements of the map. If relevant, please add the limit of 26 km on the map.

Line 124: I think it is “seawater” rather than “seaward”.

Line 132: change to Table 1 (without the).

Lines 136-138: the verb is missing in this sentence.

Lines 167-172: I am afraid I am not entirely convinced by the two-way ANOVA setting here. To me, there is a single fixed factor (ecosystem), while the station factor is random and nested in the fixed factor. Please, if you decide to retain your analysis, justify why you retained two fixed factors that are clearly related in the two-way ANOVA. Or did the Authors perform two one-way ANOVAs, instead? It is not clear.  

Results

Fig. 2. Caption: please explain the acronyms (e.g., Tw, Ts etc.).

Fig. 3. Please, add the title of x-axis and revise the caption to explain that those are stations.

Lines 206-208: these results need to be presented along with their Pseudo-F values and p-values, and subscripts. Please, consider adding this information. If you do not want to add them to the main text, you can use a Supplementary File.

Lines 208-211: these results stem from the post-hoc tests and not from PERMAMOVA as indicated in Table 2. In fact, the PERMANOVA test says that there is a difference among the three levels of the factor “ecosystem”, while the post-hoc pairwise tests say which pairs are actually different. Please, revise Table 2. The t- and p-values must be shown, along with their subscripts. Please, if you do not want to add them to the main text or Table, you can use a Supplementary File.

Table 2. Caption: please, explain all acronyms.

Figure 4: please, insert the title of Y-axis.

Line 226: please, add the subscripts (degrees of freedom) to the F variable.

Line 230: it is Nematoda. Please, revise this typo throughout the text because it occurs in other places.

Line 242: it is nematodes (no capital letter).

Line 243: please, change to Copepods and nauplii (no capital letter in Nauplii).

Figure 5: please, add the title of Y-axis.

Lines 255-259: these results are unclear because the Authors did not provide the subscripts of the Pseudo-F, which prevents fully understanding the analysis. Why “between station”? Is it rather “among stations”? Even if the PERMANOVA indicates a difference among stations, the Authors must use the post-hoc tests to assess which pairs of stations are actually diverse from each other. Please, revise.

Lines 257-259. I trust the Authors’ results, but this is not the way to present the results of a SIMPER test. Please, revise by providing all % values for all taxa and the relative p-values. You can use a Supplementary File.

Figure 6. Please, add the title of Y-axis and check out the commas in the decimals.

Lines 276-281: I trust the Authors’ results, but this is not the way to present the results of an  ANOVA test. Please, revise by providing all test values (F, p-values and subscripts) for all indices in a Table. You can use a Supplementary File.

Table 3. Please, mind the commas at the decimals. Explain the acronyms in the caption or add a sentence like this (acronyms are as in Table…), if you have explained them previously.

Lines 283-286: which is the variance explained by the marginal test? The R2 is not indicated in Table 3 for these tests.

Discussion

Line: 368: replace with “was that of Nematoda”.

Line 369: replace by [49,53].

Line 374: respond.

Lines 407-409: to support the hypothesis that copepods may be affected by nitrogen compounds, please consider the following studies:

1)  Di Lorenzo T, Fiasca B, Di Cicco M, Vaccarelli I, Tabilio Di Camillo A, Crisante S, Galassi DMP. Effectiveness of Biomass/Abundance Comparison (ABC) Models in Assessing the Response of Hyporheic Assemblages to Ammonium Contamination. Water. 2022; 14(18):2934. https://doi.org/10.3390/w14182934

2) Di Lorenzo T., Fiasca B., Di Cicco M., Cifoni M., Galassi D.M.P., 2021. Taxonomic and functional trait variation along a gradient of ammonium contamination in the hyporheic zone of a Mediterranean stream. Ecological Indicators, 132: 108268. https://doi.org/10.1016/j.ecolind.2021.108268

Reviewer 2 Report

The manuscript by Nguyen Thanh Hien and colleagues describes the densities of nematodes and rotifers at 20 sampling stations along the freshwater-estuarine ecotone of the Van Uc river in Vietnam in April 2021. The authors found that nematodes were the most abundant group at all 20 stations and that groups other than nematodes and rotifers were present in very low numbers.

While this work focusses on an important group in freshwater and marine environments - the meiofauna-, I feel the authors need to collect more data to describe the organisms in this environment. There was only one sampling occasion in April 2021, at 20 sampling stations (not sure if there was replication [methods say there was not]) and this is simply not enough data for a scientific publication because the observed patterns could be random.

I am worried that the meiofauna was fixed – this makes identification of many groups impossible (especially bdelloid rotifers, turbellarians, micro-annelids etc.). Also, the sample volume was very low, meaning you miss the larger meiofauna. Species were not identified yet many biodiversity indices were calculated – they are meaningless because we are looking at a total count of 5 different entities (‘nematoda’, ‘rotifera’ ‘nauplii’, ‘copepoda’ and ‘others’).

DETAILED COMMENTS

Abstract

I believe you must find more data and a different focus for this study.

Introduction

The introduction should have a different focus. It is cheaper and quicker to measure salt concentration rather than to look bio-indictors. You did not follow a salinization process (you have no ‘before’ and ‘after’ data) – hence loose this background. Further, species were not identified – these would in fact tell you something about the conditions in the sediment. That ratio of meiofaunal groups cannot be used for biomonitoring. The focus should be that this is the study of an ecotone and the density of nematodes.

Methods

I am concerned about the low sample size: 20 in total, sample processing (fixing the fauna- destroying many individuals of soft bodied meiofauna0 and low taxonomic resolution.

Which taxonomic keys did you use?

An awful lot of statistics are thrown at a very low sample size.

Results

You aim here should be to tell the reader what the trends and patterns are regarding the organisms, not what the PCA looks like.

You report a range of taxa here that are not mentioned in the methods not shown in a results figure but the number giving their percentage does not make any sense.

I don’t see any evidence for the nematode-rotifer pattern that you describe – there are two samples with a lot of rotifers- that is all..

Figures

Figure 2: what are the abbreviations? Explain in the figure legend.